# Allotopic expression of *COX6* elucidates Atco-driven co-assembly of cytochrome oxidase and ATP synthase

Leticia Veloso R Franco[1,2] , Chen-Hsien Su[1], Lorisa Simas Teixeira[1], Jhulia Almeida Clarck Chagas[2] , Mario Henrique Barros[2] , Alexander Tzagoloff[1]

The Cox6 subunit of *Saccharomyces cerevisiae* cytochrome oxidase (COX) and the Atp9 subunit of the ATP synthase are encoded in nuclear and mitochondrial DNA, respectively. The two proteins interact to form Atco complexes that serve as the source of Atp9 for ATP synthase assembly. To determine if Atco is also a precursor of COX, we pulse-labeled Cox6 in isolated mitochondria of a *cox6* nuclear mutant with *COX6* in mitochondrial DNA. Only a small fraction of the newly translated Cox6 was found to be present in Atco, which can explain the low concentration of COX and poor complementation of the *cox6* mutation by the allotopic gene. This and other pieces of evidence presented in this study indicate that Atco is an obligatory source of Cox6 for COX biogenesis. Together with our finding that *atp9* mutants fail to assemble COX, we propose a regulatory model in which Atco unidirectionally couples the biogenesis of COX to that of the ATP synthase to maintain a proper ratio of these two complexes of oxidative phosphorylation.

## Introduction

Cytochrome oxidase (COX), *bc1* complex, and ATP synthase of the yeast mitochondrial oxphos system are hetero-oligomeric enzymes, each composed of 11 or more polypeptides, some of which are encoded in mitochondrial and others in nuclear DNA, as reviewed elsewhere (Barros & McStay, 2020; Franco et al, 2020b). COX, responsible for the reduction of oxygen to water, consists of three catalytic core subunits (Cox1, Cox2, and Cox3), each derived from a mitochondrial gene and of 8–10 additional structural subunits that are expressed from nuclear genes. The latter are transported and sorted into the inner membrane and matrix compartments of the mitochondria (Neupert & Herrmann, 2007). Although the structural subunits do not participate directly in either the reduction of oxygen or transfer of protons to the matrix, with a few exceptions, they are essential for assembly and stability of COX (Tzagoloff & Dieckmann, 1990).

We have previously described the presence of high molecular weight complexes in yeast mitochondria, named Atco, which are composed of Cox6 and Atp9 (Su et al, 2014a; Franco et al, 2020a). In *Saccharomyces cerevisiae*, Atp9 is a mitochondrially encoded subunit that oligomerizes into a 10-subunit ring of the $F_1$-$F_o$ ATP synthase (Dautant et al, 2010), whereas in humans, Atp9 is encoded by the nucleus and forms an 8-subunit ring (Watt et al, 2010). Its rotation, promoted by protons flow through the $F_o$ portion, drives the conformational changes necessary for the synthesis of ATP by the $F_1$ portion (Boyer, 1997; Stock et al, 2000). Cox6, on the other hand, is a subunit of yeast COX that is peripherally bound to this complex on the matrix side of the inner membrane (Rathore et al, 2019).

We have shown before that Atp9 from Atco is the exclusive source for the Atp9 ring of the ATP synthase. In that study, pulse-chase experiments demonstrated that the newly translated Atp9, present in Atco and complexed to Cox6, is converted into the Atp9 ring, which is incorporated into the ATP synthase (Franco et al, 2020a). Radiolabeling of mitochondrial gene products is a powerful tool for studying the biogenesis of these hetero-oligomeric enzyme complexes. Their assembly intermediates are present in very small amounts that presently can only be detected by radiolabeling. Hence, to study the regulatory role of Atco for COX assembly, we needed the means to detect and differentiate newly translated Cox6 as a stand-alone protein and as a component of Atco. In the present study, we explored the feasibility of studying COX assembly by pulse-chase labeling of Cox6 expressed from a *COX6* gene in mitochondrial DNA (mtDNA). To this end, we used the method of mitochondrial transformation developed by Fox and colleagues (Bonnefoy & Fox, 2007). A recoded version of the nuclear *RIP1* gene for the Rieske FeS protein, an essential catalytic subunit of the *bc1* complex, was previously successfully transferred to mtDNA (Golik et al, 2003). The allotopic mitochondrial expression together with the easy radiolabeling of mitochondrial products facilitates protein sorting and fate studies. Although it is also possible to label nuclear gene products, the analysis of this class of proteins is fraught with technical problems because of the background contributed by some 6,000 cytoplasmic proteins that would also be labeled in such experiments.

---

[1]Department of Biological Sciences, Columbia University, New York, NY, USA   [2]Instituto de Ciências Biomédicas, Universidade de São Paulo, São Paulo, Brasil

Correspondence: aat3@columbia.edu

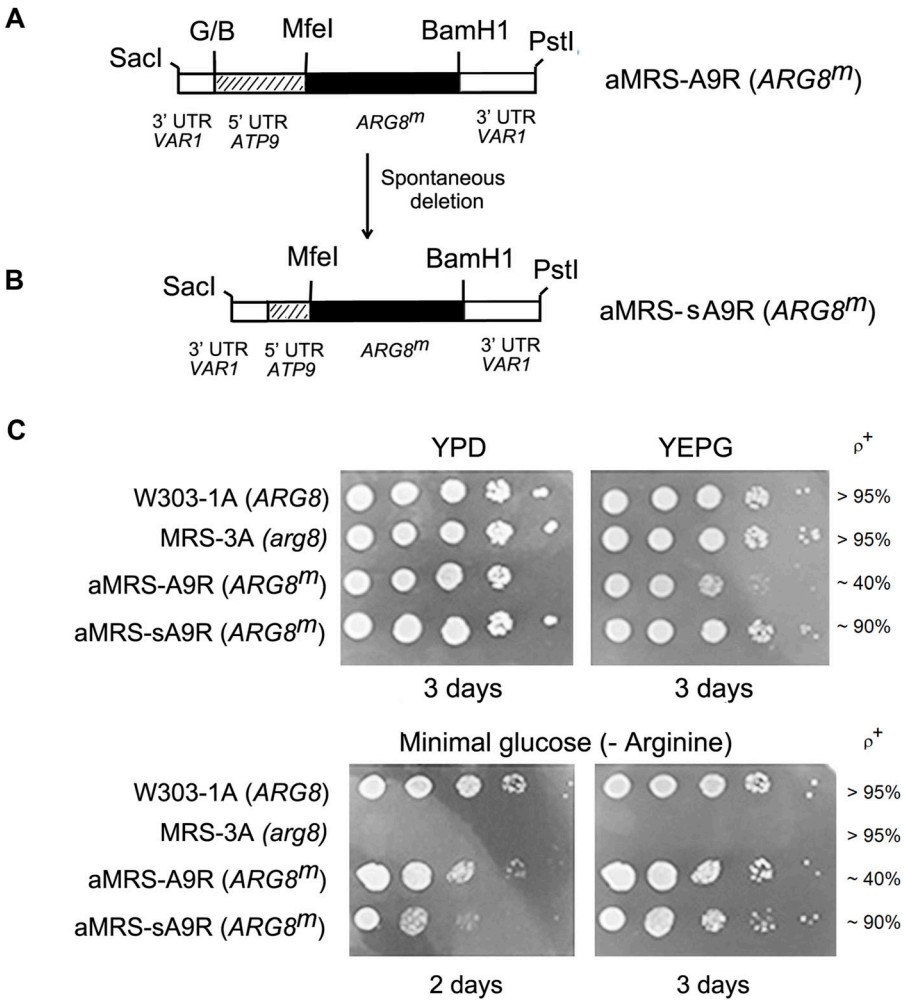

**Figure 1. Maps and expression of *ARG8^m* under the control of the *ATP9* promoter.**
**(A)** Map of *ARG8^m* fused to the 5′ UTR of *ATP9* and inserted at the BamH1 site downstream of *VAR1*. G/B indicates a ligation made from BamHI and BglII fragments with compatible ends. A complete description of the allele can be found at the Materials and Methods section. **(B)** Map of the MRS/sA9R in which part of the *ATP9* promoter region of MRS/A9R was spontaneously deleted. **(C)** Spot growth tests of the WT strain W303-1A, the *arg8* mutant MRS-3A, aMRS-A9R, an *arg8* mutant with a mitochondrial copy of *ARG8^m* under the control of the full *ATP9* promoter, and aMRS/sA9R with a partially deleted *ATP9* promoter. In the upper panels, cells grown on solid ethanol–glycerol (YPEG) were serially diluted and spotted on rich glucose YPD and on rich non-fermentable ethanol/glycerol media YPEG. On the lower panels, cells grown on solid YPEG were serially diluted and spotted on minimal glucose medium-lacking arginine. The plates were incubated for the indicated number of days at 30°C.

We present evidence that the mitochondrial version of *COX6* with a C-terminal protein C tag (*COX6-C^m*) under the transcription and translational control signals of the 5′ UTR region of *ATP9* can partially restore the ability of a *cox6* null mutant to grow on non-fermentable carbon sources. The factors impeding a complete rescue of the respiratory-deficient phenotype and the role of Atco in coupling COX biogenesis to that of ATP synthase are discussed.

## Results

### Assessment of a locus and promoter for *COX6-C^m* by using *ARG8^m*

Because *COX6* has not previously been relocated to the mitochondrial genome, we first tested the choice of the *ATP9* promoter and locus downstream of *VAR1* in mtDNA using recoded *ARG8* allele (*ARG8^m*), which is known to complement an *arg8* mutant, when its product, acetylornithine aminotransferase, is expressed from the mtDNA. A yeast strain (aMRS-A9R) in which *ARG8^m*, fused at its 5′ end to the *ATP9* promoter, was inserted at the BamHI site downstream of the mitochondrial *VAR1* gene (Fig 1A). *ARG8^m* of aMRS-A9R

at this location complemented the arginine auxotrophy of the *arg8* mutation of MRS-3A (Fig 1C, lower panels). Growth in rich ethanol glycerol (YPEG), however, was slower than that of the WT, probably because aMRS-A9R cultures consist of about 60% $\rho^{-/0}$ cells (Fig 1C, upper panels). The instability of mtDNA in aMRS-A9R was corrected by spontaneous deletion of a part of the *ATP9* promoter (Fig 1B). This new strain was named aMRS-sA9R (s stands for short), consisting of 90% $\rho^{+}$ cells, resulting in growth comparable with that of the WT in non-fermentable carbon sources (Fig 1C, upper panels).

Even though the partial deletion of the *ATP9* promoter in aMRS-sA9R stabilized mtDNA, growth on arginine-less minimal glucose medium was slower (Fig 1C, lower panel). This is expected as the partial deletion in aMRS-sA9R includes the consensus sequence 5′-ATATAAGTA-3′ in which the last A is the +1 position of the *ATP9* transcript (Dieckmann & Staples, 1994). Assays of mitochondrial translation in isolated mitochondria indicated that whereas radiolabeled $^{35}$S-methionine/cysteine is incorporated into a band corresponding to acetylornithine aminotransferase in aMRS-A9R mitochondria, the protein was barely detectable in the aMRS-sA9R mutant (Fig 2A). Unlike the severe decrease in translation of acetylornithine aminotransferase, Western analysis indicated that its

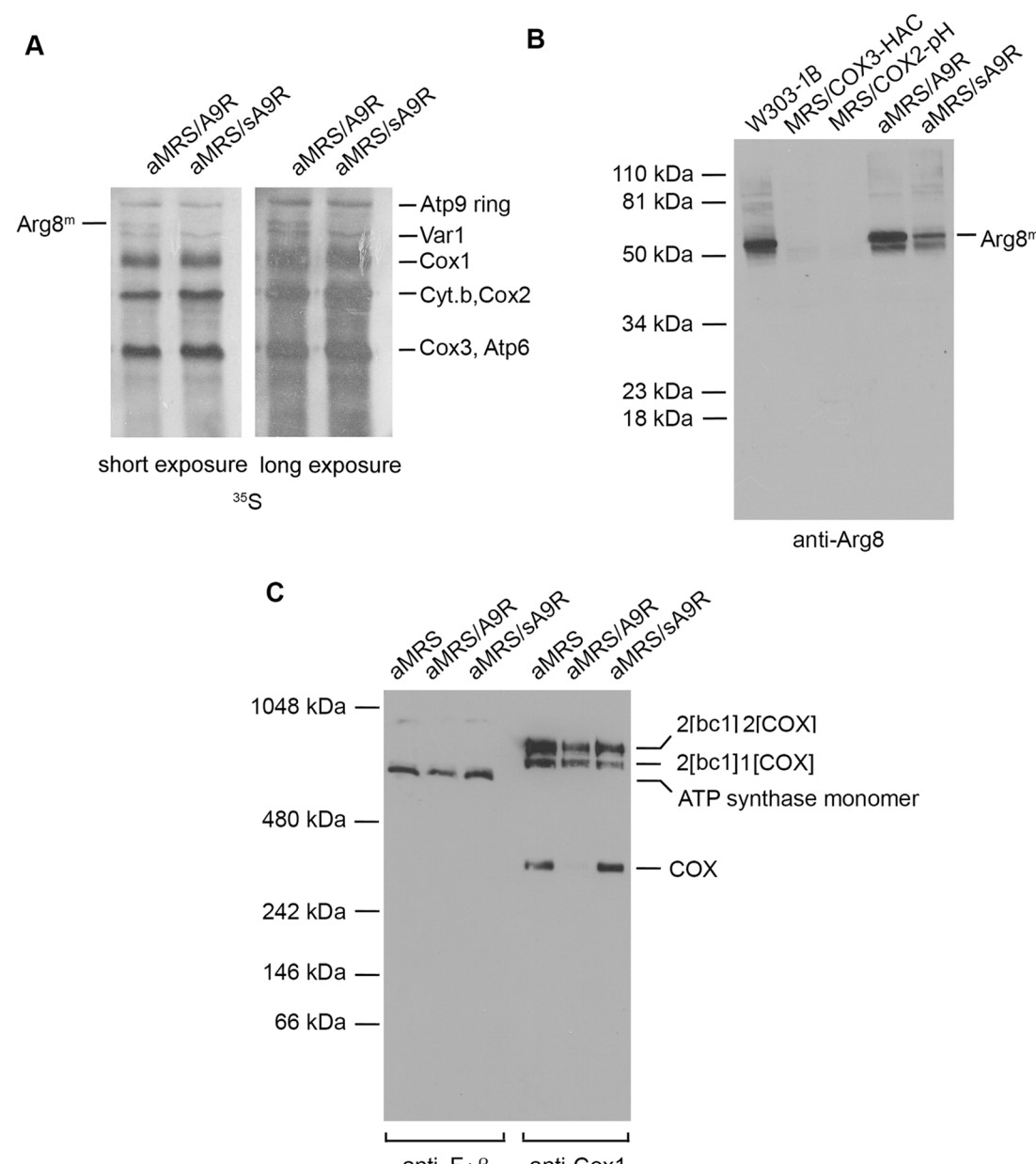

**Figure 2. Expression of *ARG8^m*.**
**(A)** aMRS-A9R and aMRS-sA9R that contained mitochondrial DNA with *ARG8^m* fused to the complete and partially deleted *ATP9* promoter region, respectively, were grown on rich liquid galactose. Mitochondria (250 μg protein) isolated from each strain were labeled with $^{35}$S-methionine/cysteine for 20 min and extracted with 2% digitonin. Proteins were separated by SDS–PAGE on a 12% polyacrylamide gel, transferred to nitrocellulose, and exposed to an X-ray film. The radiolabeled mitochondrial gene products are identified in the margin. **(B)** Mitochondria (50 μg of protein) from the WT strain W303-1B, MRS/COX3-HAC, and MRS/COX2-pH both harboring the *arg8* null mutation, and aMRS-A9R and aMRS-sA9R, were separated by SDS–PAGE in a 12% polyacrylamide gel and transferred to a nitrocellulose membrane for Western blot analysis. The blot was reacted with a primary rabbit polyclonal antibody against yeast acetylornithine aminotransferase followed by a secondary antibody against rabbit IgG conjugated to peroxidase. **(C)** Mitochondria (50 μg protein) from the *arg8* mutant strain MRS-3A and from aMRS-A9R and aMRS-sA9R were extracted with 2% digitonin and separated on a non-denaturing 4–13% polyacrylamide gel by BN–PAGE. Proteins were transferred to a PVDF membrane and reacted with a primary polyclonal antibody against the β-subunit of $F_1$ ATPase and separately against the Cox1 subunit of COX. Proteins were detected with SuperSignal chemiluminescent substrate kit (Pierce). In all experiments, aMRS-A9R was estimated to consist of 40% $\rho^+$ cells.

steady-state concentration in aMRS-sA9R was much less affected and more consistent with a slower growth phenotype of the mutant (Fig 2B).

The effect of the second copy of the *ATP9* promoter in mtDNA on the steady-state concentration of ATP synthase and COX was assessed by Western blot of the two complexes after separation by BN–PAGE. No significant difference of ATP synthase was seen between the WT, aMRS-A9R, and aMRS-sA9R. The presence of about 60% $\rho^{0/−}$ cells in vegetatively grown cultures of aMRS-A9R may account for the reduction in COX (Fig 2B and C).

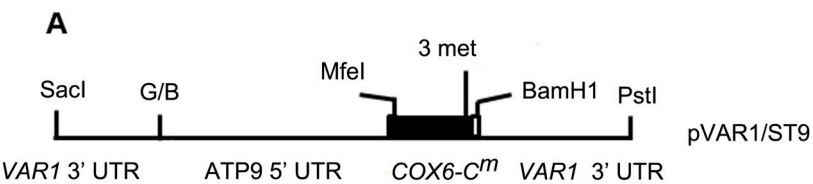

**A**

SacI · G/B · MfeI · 3 met · BamH1 · PstI · pVAR1/ST9

*VAR1* 3' UTR · ATP9 5' UTR · *COX6-C^m* · *VAR1* 3' UTR

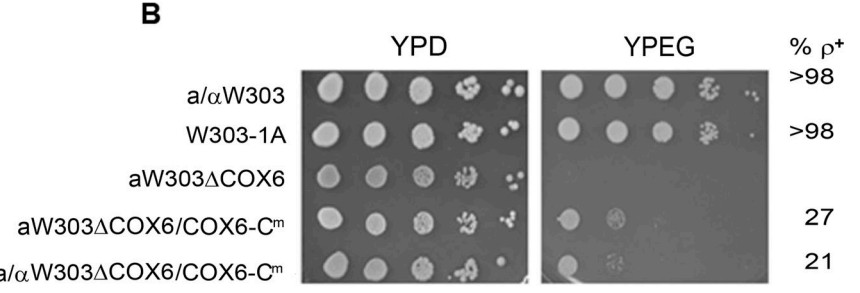

**B**

| | YPD | YPEG | % ρ⁺ |
|---|---|---|---|
| a/αW303 | | | >98 |
| W303-1A | | | >98 |
| aW303ΔCOX6 | | | |
| aW303ΔCOX6/COX6-Cᵐ | | | 27 |
| a/αW303ΔCOX6/COX6-Cᵐ | | | 21 |

**Figure 3. Complementation of *cox6* mutants by the *COX6*-Cᵐ allele.**
**(A)** Structure of the *COX6-C^m* allele. *COX6-C^m* is preceded by the promoter region of *ATP9* and ends with three methionine codons (3 met) followed by the protein C tag. G/B indicates a ligation made from BamHI and BglII fragments with compatible ends. A complete description of the allele can be found in the Materials and Methods section. **(B)** Serial dilutions of the respiratory competent diploid a/αW303 and haploid W303-1A strains and of the diploid and haploid *cox6* mutants expressing *COX6-C^m*. Cells grown on solid ethanol–glycerol were spotted on rich glucose (YPD) and rich ethanol–glycerol (YPEG) and incubated at 30°C for 3 d. The percentages of ρ⁺ cells in each culture are indicated in the right margin.

## Mitochondrially encoded Cox6 strain is respiratory competent

As $ARG8^m$ fused to the *ATP9* promoter with a location downstream of the *VAR1* locus complemented the *arg8* mutant, we proceeded with insertion of the recoded *COX6-C^m* allele under the control of *ATP9* promoter downstream of *VAR1*. The sequence of the nuclear *COX6* in this construct was modified to express a protein with Leu73 (CTC to TTG) and Leu143 (CTA to TTA) as in yeast mitochondria, the CUN codon family codes for threonine (Macino et al, 1979; Bonitz et al, 1980). In addition, the presequence was removed and three methionine codons followed by the sequence of the protein C epitope were added to the C-terminus of the protein. The hybrid allele *COX6-C^m* (Fig 3A) was introduced into the mitochondrial genome by biolistic transformation, resulting in W303ΔCOX6/COX6-C^m and a/αW303ΔCOX6/COX6-C^m, haploid and diploid strains, respectively.

*COX6-C^m* complements the *cox6* mutant as evidenced by its growth, albeit at a slower rate, on non-fermentable carbon sources (Fig 3B). The apparent partial rescue of the growth phenotype is in part contributed by the instability of mtDNA. Both the diploid and haploid strains produce about 70–80% $\rho^{-/0}$ mutants that do not respire.

To determine if the partial rescue was caused by the presence of the tag, the nuclear *COX6-C* allele, containing the three methionine codons followed by the protein C tag, was recombined into the nuclear DNA of the *cox6* mutant, and the resulting protein was visualized by SDS–PAGE (Fig S1A). This allele, when inserted into the nuclear genome, completely restored the respiratory deficiency of the mutant (Fig S1B), indicating that the tag is not responsible for the poor growth of the strains containing mitochondrial *COX6*-Cᵐ.

This led us to conclude that Cox6, when encoded in the mitochondria, leads to a respiratory competent strain, although its growth on respiratory substrates is slower than that of WT.

## Newly translated and steady-state levels of mitochondrially encoded Cox6

The partial restoration of respiratory growth of yeast with allotopic *COX6-C^m* (a/αW303ΔCOX6/COX6-C^m) encouraged us to further assess the role of Atco in COX assembly using in organello radiolabeling of mitochondrial gene products. To minimize the tendency of the strain with *COX6-C^m* to convert to $\rho^{-/0}$ mutants when grown on fermentable sugars such as glucose or galactose, mitochondria were isolated from cultures grown on a rich solid medium containing ethanol and glycerol as carbon sources to select for cells retaining full-length mtDNA. To verify the expression of *COX6-C^m*, mitochondria from WT and a/αW303ΔCOX6/COX6-C^m were labeled with $^{35}$S-methionine/cysteine. Mitochondrial proteins were extracted with 2% digitonin and purified on protein C antibody beads. Both the extracts and the fraction purified on the protein C beads were analyzed by SDS–PAGE. The fraction eluted from the protein C antibody beads displayed two labeled bands, not present in the WT control that migrated unlike any known mitochondrial gene product (Fig 4A). One of these bands migrated at a position expected for tagged Cox6. We hypothesize that the second faster migrating band corresponds to a partially degraded product. Detection of this band by in organello radiolabeling and not in Western blot can be explained by the higher sensitivity of the radiolabeling method. Importantly, mitochondrially encoded Cox6 pulled down Atp9 as was previously shown for tagged Cox6 expressed from a nuclear gene (Su et al, 2014a; Franco et al, 2020a). Total mitochondrial proteins of a/αW303ΔCOX6/COX6-C^m and of WT were also analyzed by Western blot with a primary antibody against Cox6. A band migrating like tagged Cox6 was detected in the a/αW303ΔCOX6/COX6-C^m strain confirming the expression of the relocated *COX6* (Fig 4B).

In this experiment, the somewhat lower steady-state levels of mitochondrially encoded Cox6 (Fig 4B) relative to that of WT could be because of increased turnover of the protein by mitochondrial proteases and is consistent with the presence of the smaller radiolabeled protein (Fig 4A). However, it is improbable that this marginal decrease in the steady-state levels of Cox6 could account for the severely poor growth of this strain on non-fermentable carbon sources. As discussed later, the respiratory deficiency of the strain expressing *COX6-C^m* is more likely to be caused by a defect in COX assembly.

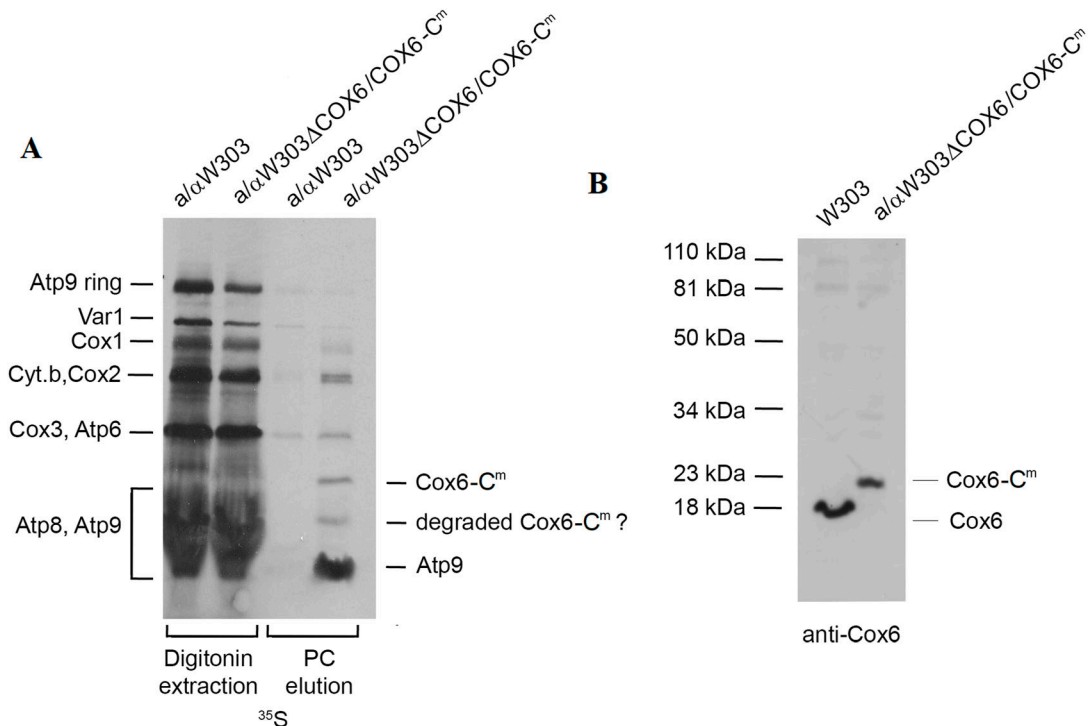

**Figure 4. Newly translated and steady-state levels of mitochondrially encoded Cox6.**
**(A)** Mitochondria (250 μg protein) of the WT diploid (a/αW303) and a/αW303ΔCOX6/COX6-C$^m$, a homozygous *cox6* mutant with a mitochondrial copy of *COX6-C$^m$* were labeled with $^{35}$S-methionine/cysteine for 20 min, extracted with 2% digitonin, and purified on protein C antibody beads. The digitonin extracts and purified fractions (PC elution) were separated by SDS–PAGE on a 12% polyacrylamide gel. Proteins were transferred to a PVDF membrane and exposed to an X-ray film. The radiolabeled mitochondrial gene products are identified in the margins. **(B)** Total mitochondrial proteins (50 μg) of WT W303 and of αW303ΔCOX6/COX6-C$^m$ were separated by SDS–PAGE on a 12% polyacrylamide gel. Proteins were transferred to a nitrocellulose membrane and blotted with a rabbit polyclonal antibody against Cox6 followed by a secondary antibody against rabbit IgG conjugated to peroxidase and visualized as in Fig 2C. Mitochondria in (A, B) were isolated from cells grown on solid rich ethanol/glycerol media.

The partial complementation by the *COX6-C$^m$* allele is similar to the partial restoration of respiration of a *rip1* null mutant with a mitochondrial-relocated *RIP$^m$* allele (Golik et al, 2003). The observed poor expression in these two studies indicates a topologic dependence in the process of sorting nuclear-encoded components into the oxphos complexes.

## Mitochondria of yeast-expressing *COX6-C$^m$* have lower steady-state levels of COX but not of the *bc1* complex or of the ATP synthase

The poor respiratory growth of the strains expressing *COX6-C$^m$* prompted us to analyze the steady-state levels of the oxphos complexes to determine whether the relocation of *COX6* had affected their assembly. To answer this question, the mitochondria of WT and of homozygous a/αW303ΔCOX6/COX6-C$^m$ were extracted with 2% digitonin and separated under non-denaturing conditions by BN–PAGE. COX, the *bc1* complex, and ATP synthase were analyzed by Western blots challenged with antibodies against Cox1 (Fig 5A), cytochrome *b* (Fig 5B), and the β subunit of F$_1$ (Fig 5C), respectively. The Western blots revealed that the supercomplexes were greatly reduced in a/αW303ΔCOX6/COX6-C$^m$ compared with WT (Fig 5A). The decrease of COX and the increase of the non-supercomplex-associated *bc1* dimer (Fig 5B) points to COX as the limiting factor in

assembly of the supercomplexes. As expected, the levels of ATP synthase in a/αW303ΔCOX6/COX6-C$^m$ were comparable with those of the WT (Fig 5C). These results led us to conclude that the slower respiratory growth of yeast with mitochondrially encoded Cox6 is because of decreased assembly of COX.

## Cox1 is the most affected catalytic subunit of COX in the strain with mitochondrially encoded Cox6

To better understand the factors responsible for the reduction of COX in the strain expressing *COX6-C$^m$*, we analyzed the steady-state levels of Cox1, Cox2, Cox3, the three mitochondrially encoded subunits of COX, and of cytochrome *b*, the only mitochondrially encoded subunit of the *bc1* complex. Total mitochondrial proteins of aW303ΔCOX6/COX6-C$^m$ were separated under denaturing conditions on a 12% polyacrylamide gel by SDS–PAGE. Whereas cytochrome *b* levels were comparable with those of WT, the COX subunits were reduced to different extents (Fig 6A and B). Interestingly, Cox1 was noticeably more affected than Cox2 and Cox3. This is consistent with previous reports showing Cox6 to be a component of the Cox1 assembly module (Mick et al, 2010; McStay et al, 2013) and also confirms that the slower respiratory growth of this strain is most likely the result of a partial block of COX assembly.

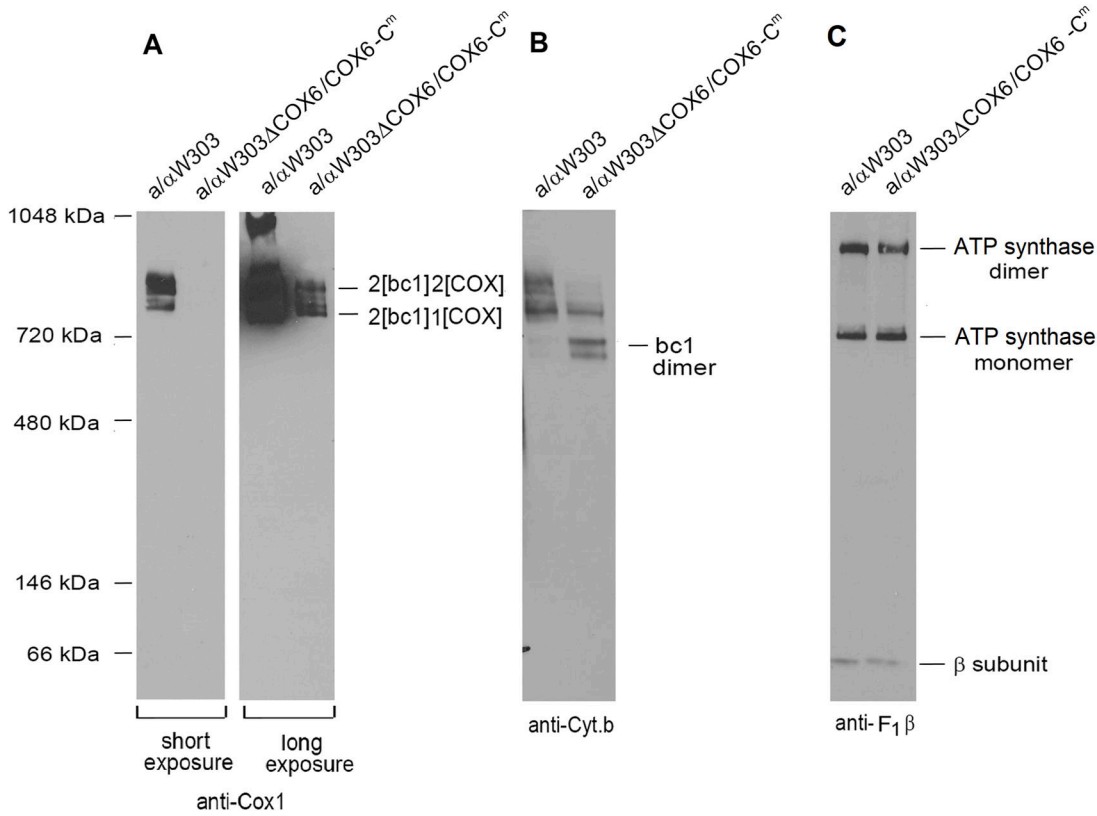

**Figure 5.  Steady-state levels of the oxphos complexes.**
Mitochondria (50 μg protein) of the WT W303-1B and of a/αW303ΔCOX6/COX6-C^m were isolated from cells grown on solid ethanol/glycerol media. The mitochondria were extracted with 2% digitonin and separated on a non-denaturing 4–13% polyacrylamide gel by BN–PAGE. **(A, B, C)** Proteins were transferred to a PVDF membrane and reacted with primary antibodies against Cox1 (A), cytochrome *b* (B), and the F₁ β subunit of the ATP synthase (C). Proteins were visualized with a secondary antibody conjugated to peroxidase as in Fig 2C.

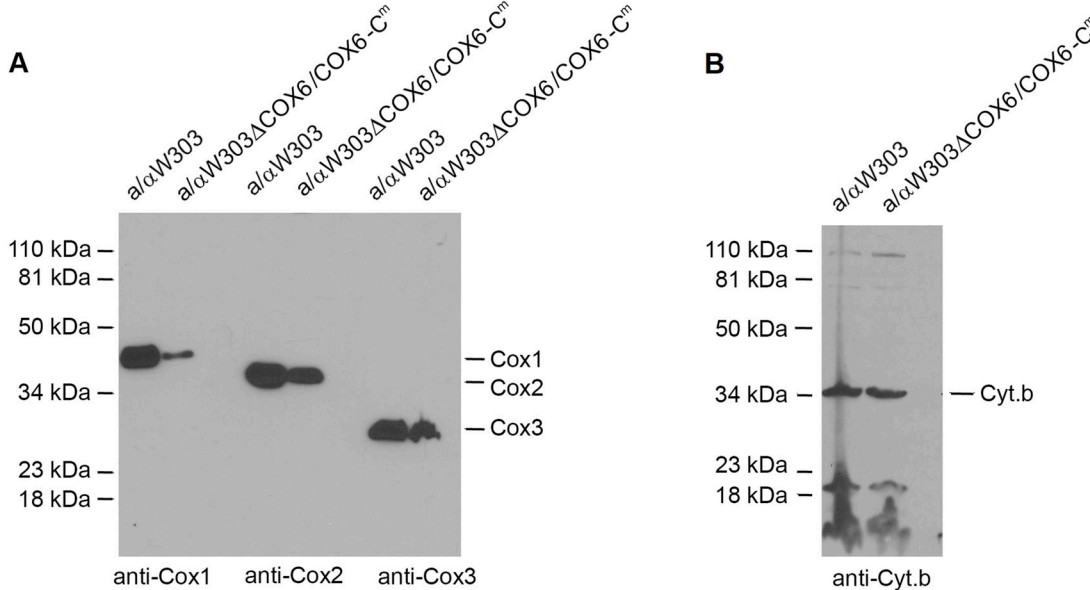

**Figure 6.  Steady-state levels of the mitochondrially encoded subunits of the respiratory chain.**
Mitochondria (50 μg protein) of the WT W303-1B and of αW303ΔCOX6/COX6-C^m were isolated from cells grown on solid rich ethanol/glycerol media. The mitochondria were separated on a denaturing 12% polyacrylamide gel by SDS–PAGE. **(A)** Proteins were transferred to a nitrocellulose membrane and reacted with primary monoclonal mouse antibodies against Cox1, Cox2, and Cox3 followed by a secondary antibody against mouse IgG conjugated to peroxidase (A). **(B)** Cytochrome *b* was detected with a primary polyclonal rabbit antibody against cytochrome *b* followed by a secondary antibody against rabbit IgG conjugated to peroxidase. Proteins were visualized as in 2C.

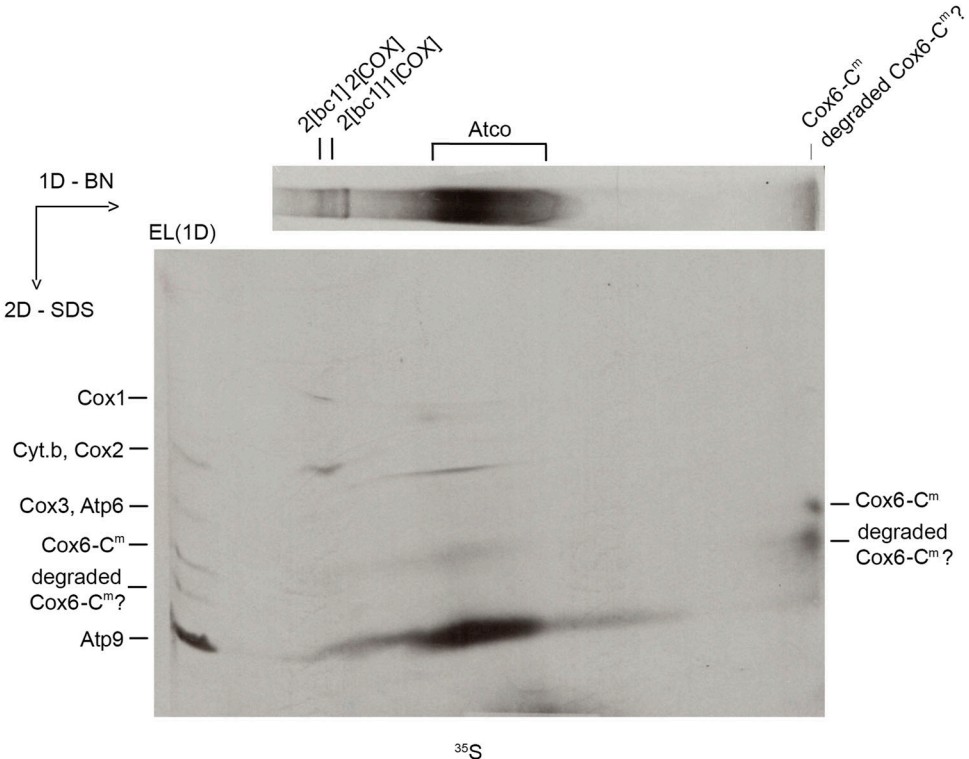

**Figure 7. Analysis of Atco and Cox6 by two-dimensional electrophoresis.**
Digitonin extracts of aW303ΔCOX6/COX6-C$^m$ mitochondria (250 μg protein) isolated from cells grown on solid ethanol/glycerol media were labeled with $^{35}$S-methionine/cysteine, purified on protein C beads, and separated on the first dimension in a non-denaturing 4–13% polyacrylamide gel by BN–PAGE followed by a second dimension on a 12% polyacrylamide gel by SDS–PAGE. Proteins were transferred to a PVDF membrane and exposed to the X-ray film. A sample of the fraction eluted from the protein C beads (EL 1D) was separated directly on the SDS gel.

### Only a small fraction of the mitochondrially encoded Cox6 is associated with Atp9 in Atco

To gain a better understanding of the reason for the severe depletion of Cox1, a digitonin extract of aW303ΔCOX6/COX6-C$^m$ mitochondria that had been labeled and purified on protein C antibody beads was analyzed by 2D electrophoresis: BN–PAGE in the first dimension followed by SDS–PAGE in the second. Atco and the supercomplexes were previously found to be pulled down in a similar experiment but using a strain in which tagged Cox6 is expressed from a nuclear gene (Su et al, 2014a; Franco et al, 2020a). The autoradiograph revealed co-purification of supercomplexes, and of a diffuse band corresponding to Atco-containing radiolabeled Atp9 and tagged Cox6 when the latter is expressed from the mitochondrial *COX6-C$^m$* gene (Fig 7). In addition, the digitonin extract of aW303ΔCOX6/COX6-C$^m$ mitochondria contained a low-molecular weight radiolabeled band near the bottom of the native gel (Fig 7). Under denaturing conditions in the second dimension, this band was resolved into two bands that migrated like mitochondrially encoded Cox6 and its hypothetical proteolytic fragment (Figs 4A and 7). Most of Cox6 and its presumed proteolytic fragment were present as a monomeric proteins, indicating that only a small fraction of mitochondrially encoded Cox6 is incorporated into Atco. Neither of these bands was present in previous pull-down experiments with the tagged Cox6 encoded in the nucleus (Su et al, 2014a; Franco et al, 2020a).

These results indicate that most of the newly translated Cox6 expressed from the mitochondrial *COX6-C$^m$* is present as a mixture of the free-standing monomeric subunit and of the shorter product proposed to be a proteolytic product. This can explain the very partial assembly of COX in aW303ΔCOX6/COX6-C$^m$ if only Cox6 of Atco is competent in entering the COX assembly pathway. These results do not, however, exclude the possibility that the free monomeric Cox6 subunit can also be incorporated into COX.

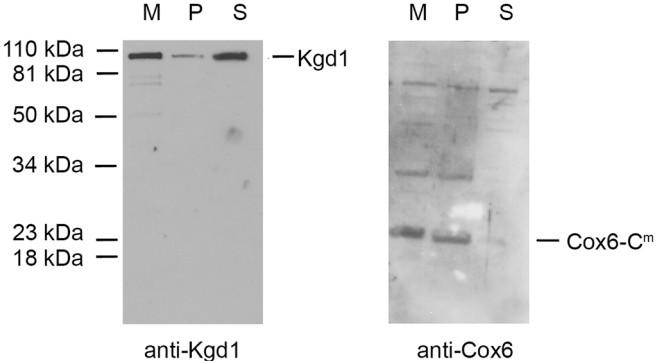

**Figure 8. Intra-mitochondrial localization of mitochondrially encoded Cox6.**
Mitochondria of aW303ΔCOX6/COX6-C$^m$ were isolated from cells grown on solid rich ethanol/glycerol media. Mitochondria (M) were sonicated and centrifuged at 105,000$g$ to separate the soluble matrix proteins (S) and inverted submitochondrial particles (P). The distribution of mitochondrially encoded Cox6 in the different fractions was assessed using an anti-Cox6 antibody. Mitochondrial breakage was checked using an antibody against the soluble matrix Kgd1.

### Localization of mitochondrially encoded Cox6 in the inner membrane

The presence of most of the mitochondrially encoded Cox6 as a free subunit made it of interest to investigate if it was a soluble matrix protein, or

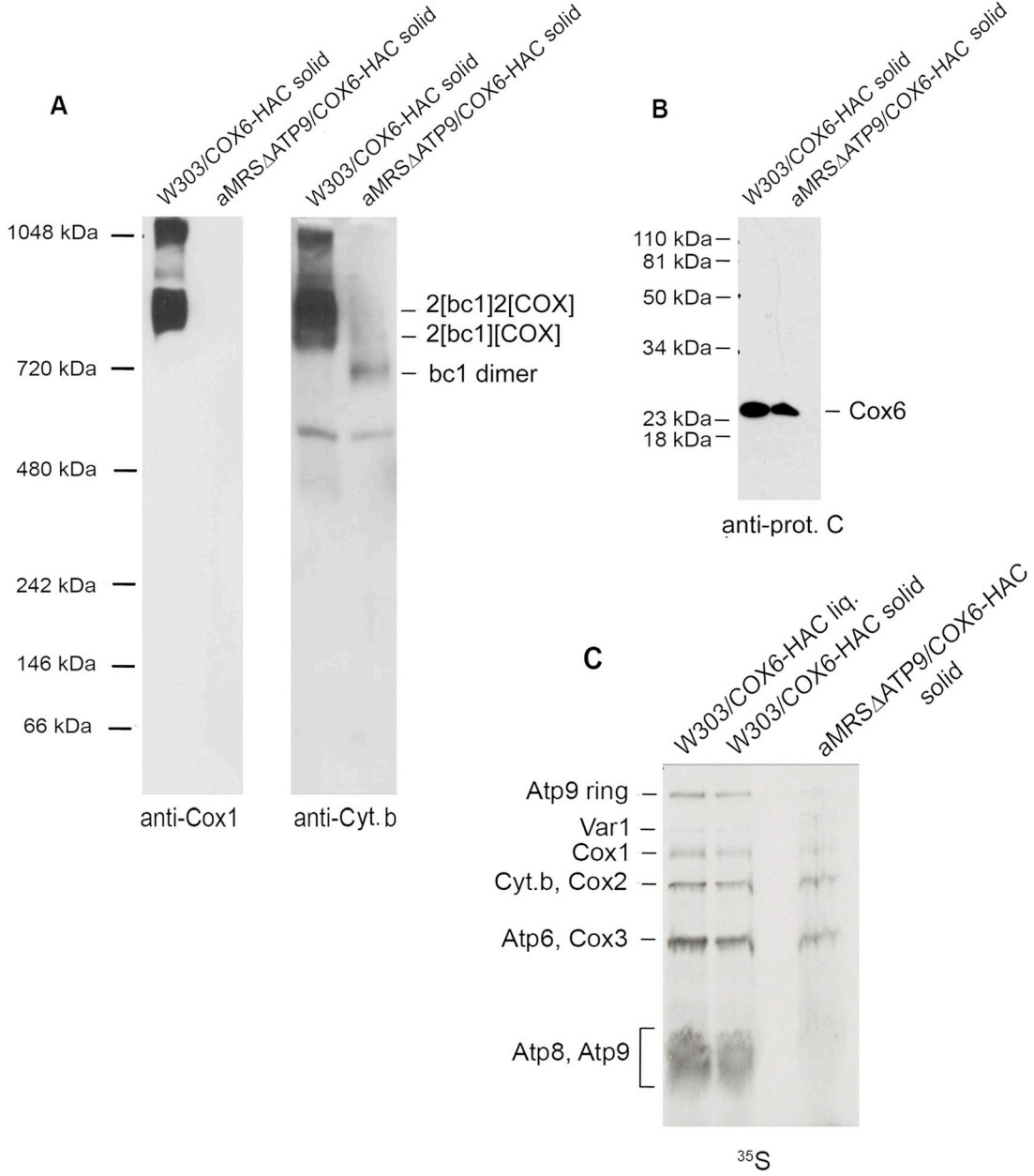

**Figure 9. Analysis of the mitochondrial gene products and supercomplexes in the _atp9_ null mutant.**
**(A)** Mitochondria were isolated from cultures of W303/COX6-HAC, a strain expressing Cox6 tagged with hemagglutinin followed by a protein C epitope from the nuclear _COX6-HAC_ gene, and of the _atp9_ null mutant aMRSΔATP9/COX6-HAC. Both strains were grown on solid rich galactose media. Mitochondria were extracted with 2% digitonin and separated under non-denaturing conditions by BN–PAGE on a 4–13% polyacrylamide gel. Proteins were transferred to PVDF membranes and reacted with the indicated primary mouse antibody against Cox1 and separately with an antibody against cytochrome b. Proteins were visualized after a reaction with a secondary antibody conjugated to peroxidase and visualized as in Fig 2C. **(B)** Total mitochondria (50 µg proteins) of W303/COX6-HAC and aMRSΔATP9/COX6-HAC, both grown on solid rich galactose medium, were separated by SDS–PAGE. Proteins were transferred to a nitrocellulose membrane and Cox6 was visualized by reacting the membrane with a primary rabbit antibody against protein C, followed by a secondary antibody against rabbit IgG conjugated to peroxidase and visualized as in 2C. **(C)** Mitochondria (250 µg protein) from W303/COX6-HAC grown either on liquid of solid rich galactose media, and from aMRSΔATP9/COX6-HAC, grown on solid rich galactose medium, were labeled with $^{35}$S- methionine/cysteine for 20 min, extracted with 2% digitonin, and separated by SDS–PAGE on a 12% polyacrylamide gel. Proteins were transferred to nitrocellulose and the blot exposed to an X-ray film. The radiolabeled mitochondrial gene products are identified in the margins. In all experiments, aMRSΔATP9/COX6-HAC was estimated to consist of 50% $p^+$ cells.

was associated with the inner membrane like the hydrophobic Atp9 ring. To localize Cox6, the mitochondria were sonicated and separated by centrifugation to obtain inside-out submitochondrial particles that are recovered in the pellet fraction and soluble matrix proteins, which under these conditions stay in the supernatant. As shown in Fig 8, mitochondrially encoded Cox6 is associated with the inner membrane, even though most of

it is not complexed to Atp9 or part of COX (Fig 7). This suggests that most of Cox6 translated on inner membrane-bound ribosomes and inserted into the membrane is incompetent to be assembled into Atco and COX.

### Atp9 is necessary for COX assembly

COX mutants, including the *cox6* mutant, contain normal amounts of ATP synthase subunits (Su et al, 2014a), indicating that Atco is not necessary for ATP synthase assembly. This suggests a unidirectional regulatory mechanism that ensures assembly of ATP synthase even in cells that do not respire to permit an ATP-dependent maintenance of membrane potential by the ATP synthase. The dependence of COX assembly on Atco was studied in the *atp9* null mutant that was grown on a solid rich galactose medium. By first pre-growing the *apt9* mutant on minimal medium lacking arginine to select for $\rho^+$ cells after subsequent growth on solid rich galactose, we were able to obtain cultures consisting of about 50% cells with full-length mtDNA, which is evidenced by in organello labeling of mitochondria (Fig 9C). The steady-state concentration of COX, mainly present in the supercomplexes, was reduced to levels not detectable by Westerns of mitochondria extracted with digitonin and separated on native gels (Fig 9A, left panel). In contrast, it was possible to detect free *bc1* dimer (Fig 9A, right panel) at levels consistent with the percentage of $\rho^+$ cells in the culture used to isolate the mitochondria. We have also checked the steady-state levels of Cox6 in the *atp9* mutant, which are comparable with those of the control strain (Fig 9B), indicating that the absence of COX in this strain is not caused by a relevant decreased supply of Cox6. These results indicate that COX assembly is dependent on the ATP synthase even though the opposite is not true.

## Discussion

Previous experiments showed that the ATP synthase is assembled in the *cox6* null mutant of *S. cerevisiae* (Su et al, 2014a). Despite not being essential for the Atp9 ring formation, Atco enhances the efficiency of the process as evidenced by the slower Atp9 ring formation in the *cox6* null mutant in pulse-chase experiments (Su et al, 2014a). These results indicated that although assembly of ATP synthase is more efficient in cells containing Atco, mitochondria can also use non-Atco Atp9 for this process.

The studies reported here constitute strong evidence that in contrast to the biogenesis of ATP synthase, biogenesis of COX is necessarily coupled to Cox6 of the Atco complex. The evidence may be summarized as follows. (1) The allotopic *COX6-C*$^m$ gene expresses nearly normal amounts of Cox6. However, *cox6* mutants harboring *COX6-C*$^m$ have very low amounts of COX, and as a result, grow poorly on non-fermentable carbon sources (Figs 3 and 5). (2) Despite nearly normal concentration of mitochondrially encoded Cox6 (Fig 4), its expression leads to severely decreased levels of Cox1, but not Cox2 or Cox3 (Fig 6). This phenotype is a hallmark of mutants defective in assembling the Cox1 module of COX. As Cox6 interacts exclusively with the Cox1 assembly intermediate, these results imply that translation of Cox6 on mitochondrial membrane-bound ribosomes renders this subunit highly inefficient in entering

the COX assembly pathway. (3) Newly translated Cox6 in the strain with mitochondrial *COX6-C*$^m$ is present predominantly as the free subunit by itself and what we proposed to be a proteolytic product (Fig 7). The putative proteolytic fragment is detected by pulse labeling but not by Western probing of the mitochondria, indicating that it is unstable and is further degraded. (4) Translation of Cox6 on mitochondrial ribosomes is coupled to its insertion into the inner membrane (Fig 8). The severely depressed level of COX suggests that the membrane location, or some other factor, prevents mitochondrially encoded Cox6 from being efficiently incorporated into the Cox1 module resulting in its proteolytic loss. (5) The inability of non-Atco Cox6 to be recruited for COX biogenesis is also supported by the results obtained with an *atp9* null mutant expressing tagged Cox6 from a nuclear gene (Fig 9). This mutant, lacking Atco, shows the total absence of COX even though the steady-state concentration of Cox6 is comparable with WT (Fig 9B). The failure to assemble COX in this strain cannot be ascribed to the loss of mtDNA as 50% of the culture used in this experiment consisted of $\rho^+$ that lacked Atp9 and Atp8 but not of the other mitochondrial gene products, which were reduced by approximately twofold, consistent with the loss of mtDNA in only 50% of the cells (Fig 9C). This is also supported by BN–PAGE analysis of digitonin extracts, which also shows the absence of the supercomplexes, but not *bc1* complex in the *atp9* mutant (Fig 9A).

These results, illustrated diagrammatically in Fig 10A–D, lead us to conclude that by being a dual source of Atp9 for ATP synthase and Cox6 for COX, Atco coordinates their biogenesis and maintains a constant ratio optimal for the ATP-generating function of the oxphos pathway. The unidirectional nature of this regulation is based on the finding that biogenesis of COX is necessarily coupled to the presence of Cox6 in Atco, whereas both free and Atco-associated Atp9 can be recruited for ATP synthase biogenesis. The mechanistic dependence of COX biogenesis has also been noted in earlier studies of *S. cerevisiae* mutants lacking Atp6 (Rak et al, 2007) or subunits e and g, each of which exhibit decreased levels of COX and an altered stoichiometry of supercomplexes (Saddar et al, 2008).

In addition to the transfer of most of their genes to the nucleus, mitochondrial evolution also entailed the acquisition of new nuclear gene products coding for constituent subunits of the respiratory complexes and the ATP synthase. These proteins, which are absent in the bacterial homologs, do not appear to have a catalytic function but nonetheless are essential for biogenesis of the enzymes. Cox6 is one of such supernumerary subunits of the mitochondrial COX respiratory complex. The contribution by the mitochondria and nuclear genomes of polypeptides destined to become part of the same hetero-oligmeric enzyme demanded the means of regulating a timely and proportionate expression of the compartmentally separated gene products. Several mechanisms have been described, in which the product of nuclear DNA either regulates mitochondrial translation of a subunit or impinges on a rate-limiting step in the assembly pathway of the oxphos complexes (Perez-Martinez et al, 2009; Rak & Tzagoloff, 2009; Gruschke et al, 2012). The dual genetic origin of these complexes necessitated other mechanisms for adjusting their ratio depending on the carbon source and other metabolic requirements. Growth of yeast under anaerobic conditions or on glucose as the sole carbon

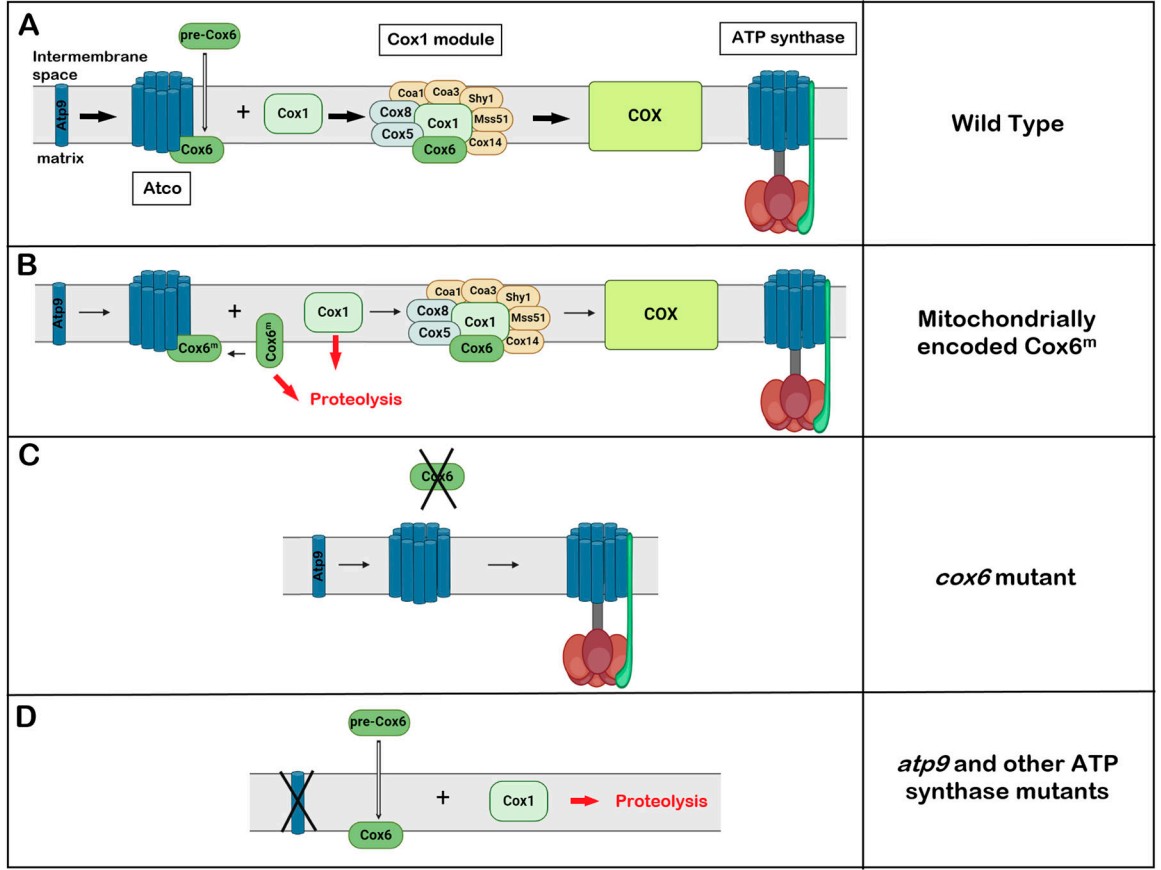

**Figure 10. Model of Atco in coupling the biogenesis of cytochrome oxidase (COX) to that of the ATP synthase.**
**(A)** In WT, Cox6 precursor synthesized on cytoplasmic ribosomes is transported (open arrow) into the mitochondrial inner membrane where it interacts with oligomeric Atp9 to form Atco. The interactions of the Atp9 in the oligomer are similar to those in the Atp9 ring (Franco et al, 2020a), but at present, it is not known if the Atp9 of Atco is present as a 10-membered ring as shown in the diagram or some other oligomeric form. Atco provides Cox6 for the Cox1 assembly module of COX, and the Atp9 oligomer to the ATP synthase assembly pathway (shown above). Not all the subunits of the membrane unit ($F_0$) are shown in the diagram of the ATP synthase. **(B)** Co-translational membrane insertion of Cox6 from mitochondrial ribosomes bound to the matrix side of the inner membrane allows some Cox6 to enter the normal COX assembly pathway. Because of the less efficient COX assembly, some of the Cox6 that failed to be incorporated into Atco is proteolytically cleaved into a smaller fragment. The retarded rate of Cox6 entry into the normal COX assembly pathway elicits substantial turnover of Cox1, but not Cox2 or Cox3. **(C)** In the *cox6* mutant, COX is not assembled. However, the Atp9 monomer synthesized on mitochondrial ribosomes can assemble into a ring and be recruited for ATP synthase assembly. **(D)** In the *atp9* mutant, cytoplasmically translated Cox6 is transport in the matrix as shown in the figure or into the inner membrane. The free non-Atco-associated subunit does not interact with the Cox1 intermediate resulting in proteolysis of the Cox1 causing the complete loss of COX.

source, for example, dramatically suppresses biogenesis of the respiratory chain, but much less so of the ATP synthase (Tzagoloff et al, 1973; Trawick et al, 1989; Wright et al, 1989). This is achieved by the HAP complex, which globally down-regulates transcription of nuclear genes that code for subunits of the respiratory chain such as *COX6* (Hahn & Guarente, 1988). The presence of ATP synthase in the *cox6* mutant is not surprising as other respiratory-deficient mutants grown under glucose-repressed conditions also contain ATP synthase needed for maintenance of a membrane potential, generated by hydrolysis of ATP. As a result, glucose-repressed cells have an altered stoichiometry of the respiratory complexes relative to the ATP synthase. We propose that the dependence of COX biogenesis on Atco-associated Cox6 is an example of how mitochondria coordinate biogenesis of this important component of the respiratory chain to that of the ATP-generating ATP synthase.

Cox6 is a hydrophilic protein that interacts with its partner subunits peripherally on the matrix site of the inner membrane (Hartley et al, 2019). Translation of Cox6 on cytoplasmic ribosomes necessitates its transport into the mitochondria to enter the COX assembly pathway. Very little is known about the routes by which the structural subunits of COX are imported into the mitochondria. A few studies have examined the sorting pathways for subunits Cox4 and Cox5a during their import (Gärtner et al, 1995; Böttinger et al, 2013). Both require matrix HSP70 for the ATP-dependent translocation of the unfolded subunits through the TIM complex of the inner membrane. In the case of Cox4, it is presumed that the protein is completely translocated into the matrix, whereas translocation of the membrane spanning Cox5a subunit is arrested by its internal hydrophobic stop-transfer domain followed by laterally sorting in the inner membrane (Gärtner et al, 1995).

At present, information regarding the import and sorting of Cox6 is lacking.

As mentioned already above, the steady-state concentration of mitochondrial-encoded Cox6 is only marginally lower than that of WT, indicating a sufficiency of Cox6 for COX assembly. At present, therefore, we favor defective sorting as the more likely explanation for the compromised COX assembly when *COX6-C^m* lacking the complete presequence is translated internally on mitochondrial ribosomes. One possible explanation is that Cox6 would normally interact with its partner subunits without completely penetrating into the matrix. Alternatively, the Cox6 precursor may undergo tandem processing of its N-terminal 40-amino acid-long presequence. In the latter scenario, the Cox6 precursor is targeted to the matrix by removal of only part of the N-terminal presequence. The remaining part of the presequence then directs Cox6 back to the inner membrane where, by removal of the remaining sorting sequence, Cox6 is rendered competent to interact with the Cox1 intermediate. Either of the two mechanisms would be blocked in our construct because of the absence of the normal Cox6 presequence or its normal import and sorting singal. The full restoration of respiration with Cox6 expressed from the nuclear *COX6-C* gene suggests that the presence of three methionine codons followed by a protein C tag at its C-terminus does not interfere with COX assembly. However, we cannot exclude that the assembly defect may also be caused by the presence of three first codons of *ATP9* at the N-terminus of *COX6-C^m*.

Translation of Cox1, 2, and 3 in the *COX6-C^m* strain, measured by pulse-labeling of isolated mitochondria, were only marginally reduced compared with WT (Fig 4A). Western analysis of these three subunits in the steady state, however, indicated substantially less Cox1 than Cox2 and Cox3 (Fig 6A). This phenotype is similar to that reported for *cox14* mutants (Barrientos et al, 2004). Cox1 translation and assembly were previously shown to be dependent on the Cox1-specific translation activator Mss51 (Perez-Martinez et al, 2003; Barrientos et al, 2004), which together with Cox14 and Coa3 forms a complex with newly synthesized Cox1. Accordingly, a new round of Cox1 translation is blocked until Mss51 is displaced from Cox1 at some yet to be defined downstream step of the COX assembly pathway (Barrientos et al, 2004; Perez-Martinez et al, 2009; Mick et al, 2010; Fontanesi et al, 2011). Translation of Cox1 in mutants lacking Cox14 or Coa3 is explained by the instability of the Cox1–Mss51 interaction in the absence of either one of these factors (Barrientos et al, 2004; Fontanesi et al, 2011). The correspondence in the phenotype of the strains expressing *COX6-C^m* and of the *cox14* and *coa3* mutants, namely their ability to translate Cox1, 2, and 3, but not to assemble COX suggests that Cox6, a constituent of the Cox1 module (McStay et al, 2013), may also function in stabilizing the Cox1–Mss51 interaction.

# Materials and Methods

### Yeast strains and grow media

The strains of *S. cerevisiae* and their genotypes are listed in Table 1. PCR primers are listed in Table 2. The compositions of the growth media were YPD, 2% glucose, 2% peptone, 1% yeast extract; YPGal, 2% galactose, 2% peptone, 1% yeast extract; YPEG, 3% glycerol, 2% ethanol, 2% peptone, 1% yeast extract; minimal glucose, 2% glucose, 0.7% nitrogen base, and auxotrophic requirements. Solid media contained 2% agar.

### Preparation and labeling of mitochondria and purification of tagged proteins

The mitochondria were prepared by the method of Herrmann et al (1994) from yeast grown at 30°C to the early stationary phase in either liquid or solid YPGal or solid YPEG. Small aliquots of mitochondria were frozen in liquid nitrogen and stored at −80°C. Mitochondria were labeled for 20 min with $^{35}$S-methionine/cysteine (3,000 Ci/mmol). The reaction was stopped with puromycin plus excess unlabeled methionine and further incubated for an additional 10 min. Digitonin extracts of the labeled mitochondria were purified on protein C antibody beads (McStay et al, 2013) and analyzed by SDS–PAGE (Laemmli, 1970) and BN–PAGE (Wittig et al, 2006).

### Construction of *ARG8^m* under the control of the *ATP9* promoter

To express *ARG8^m* in a respiratory competent background, we constructed a plasmid in which the gene, fused to the promoter region of mitochondrial *ATP9* gene at its 5′ end, was inserted into an AT-rich region downstream of *VAR1*. In this construct, *ARG8^m* is preceded by the first three codons of *ATP9*. The *VAR1* 3′-UTR sequence consisting of 870 nucleotides that included the BamHI site was amplified from W303 mtDNA with primers var1–1 and var1–2 and was cloned into a modified YIp349, a yeast/*E. coli* integrative shuttle vector in which the *URA3* marker of YIp352 (Hill et al, 1986) was replaced with a 1-kb fragment containing *TRP1*. The resulting plasmid (pVAR1/ST1) was linearized at the BamHI site approximately midway in the 3′ UTR sequence and was ligated to a BglII–BamHI fragment containing the 5′ sequence of A*TP9*, which had been amplified with primers var1–3 and var1–5. The resulting plasmid (pVAR1/ST7) was digested with MfeI and BamHI and ligated to a 308-bp MfeI–BamHI fragment containing the C-terminal region of *ARG8^m*. This fragment was obtained from *ARG8^m* that had been amplified with primers arg8m-2 and arg8m-12. To reconstitute *ARG8^m*, the resultant plasmid, pVAR1/ST10, was digested with MfeI and ligated to a 965-bp long fragment containing the N-terminal region of *ARG8^m*. This ligation yielded pVAR1/ST11 with the insert shown in Fig 1A. The insert of pVAR1/ST11 was transferred to pJM2 (Mulero & Fox, 1993) in which the PstI site on the 3′ side of *COX2* had been destroyed. The resultant plasmid, pVAR1/ST12 was introduced into the mitochondria of the *kar1* mutant DFKρ$^0$ by biolistic transformation (Bonnefoy & Fox, 2007). The presence of the *ARG8^m* allele in mtDNA was tested by a cross to the *cox2* tester aM9-94/A1 and by checking the diploid cells issued from the cross for respiratory competence. The synthetic petite DFK/VAR1/ST12ρ$^−$ obtained from the transformation was crossed to MRS-3A, a strain that harbors an *arg8* null mutation. Recombinants were selected by their arginine prototrophy, yielding strain aMRS-A9R.

**Table 1.** Genotypes and sources of the *S. cerevisiae* strains used in this study.

| Strain | Relevant genotype | mt DNA | Source |
|---|---|---|---|
| W303-1A | MATa leu2-3,112 trp1-1 ura3-1 ade2-1 his3-11,15 | ρ⁺ | R Rothstein, Columbia University |
| W303-1B | MATα leu2-3,112 trp1-1 ura3-1 ade2-1 his3-11,15 | ρ⁺ | R Rothstein, Columbia University |
| MR6 | MATa ade2-1 his3-11,15 trp1-1 leu2-3,112 ura3-1 Δarg8::HIS3 | ρ⁺ | Rak et al (2007) |
| MRS-3A | MATa leu2-3,112 trp1-1 ura3-1 ade2-1 his3-11,15 Δarg8::HIS3 | ρ⁺ | McStay et al (2013) |
| MRS-3B | MATα leu2-3,112 trp1-1 ura3-1 ade2-1 his3-11,15 Δarg8::HIS3 | ρ⁺ | McStay et al (2013) |
| aM9-94/4B | MATa ade1 | cox2 | Coruzzi and Tzagoloff (1979) |
| M9-94/A1 | MATα met | cox2 | Coruzzi and Tzagoloff (1980) |
| MRS/COX3-HAC | MATα leu2-3,112 trp1-1 ura3-1 ade2-1 his3-11,15 Δarg8::HIS3 | ρ⁺ COX3-HAC[a] | Su et al (2014b) |
| MRS/COX2-pH | MATα leu2-3,112 trp1-1 ura3-1 ade2-1 his3-11,15 Δarg8::HIS3 | ρ⁺ COX2-pH[b] | Franco et al (2018) |
| DFKρ⁰ | MATα kar1-1 ade2-101 leu2D ura3-52 lys2 Δarg8::URA3 | ρ⁰ | McStay et al (2013) |
| DFK/VAR1/ST12ρ⁻ | MATα kar1-1 ade2-101 leu2D ura3-52 lys2 Δarg8::URA3 | ρ⁻ 3'VAR1::ARG8ᵐ | This study[c] |
| aMRS/VAR1/ST12ρ⁺ = aMRS-A9R | MATa ade2-1 his3-1,15 leu2-3,112 trp1-1 ura3-1 Δarg8::HIS3 | ρ⁺ 3'VAR1::ARG8ᵐ | This study[c] |
| aMRS/VAR1/ST13ρ⁺ = aMRS-sA9R | MATa ade2-1 his3-1,15 leu2-3,112 trp1-1 ura3-1 Δarg8::HIS3 | ρ⁺ 3'VAR1::ARG8ᵐ | This study[d] |
| DFK/VAR1/ST9ρ⁻ | MATα kar1-1 ade2-101 leu2D ura3-52 lys2 Δarg8::URA3 | ρ⁻ 3'VAR1::COX6-Cᵐ | This study[e] |
| aW303ΔCOX6 | MATa ade2-1 his3-1,15 leu2-3,112 trp1-1 ura3-1Δ cox6::URA3 | ρ⁺ | Glerum and Tzagoloff (1997) |
| aW | MATa ade2-1 his3-1,15 leu2-3,112 trp1-1 ura3-1 Δcox6::URA3 | ρ⁰ | This study |
| a/αW303 ΔCOX6/COX6-Cᵐ | MATa,α ade2-1/ade2-1 his3-1,15/his3-1,15 leu2-3,112/leu2-3,112 trp1-1/trp1-1 ura3-1 Δcox6::URA3/Δcox6::URA3 | ρ⁺ 3'VAR1:: COX6-Cᵐ | This study |
| aW303 ΔCOX6/COX6-Cᵐ | MATa ade2-1 his3-1,15 leu2-3,112 trp1-1 ura3-1 Δcox6::URA3 | ρ⁺ 3'VAR1:: COX6-Cᵐ | This study |
| W303/COX6-HAC | MATα ade2-1 his3-1,15 leu2-3,112 trp1-1 ura3-1 Δcox6::URA3 trp1::pG71/ST9[f] | ρ⁺ | McStay et al (2013) |
| W303/COX6-C | MATα ade2-1 his3-1,15 leu2-3,112 trp1-1 ura3-1 Δcox6::URA3 trp1::YIp349-COX6-3met-C#7[g] | ρ⁺ | This study |
| aMRSΔATP9/COX6-HAC | MATa ade2-1 his3-11,15 leu2-3,112 trp1-1 ura3-1 Δcox6::URA3 trp1::pG71/ST9[f] Δarg8::HIS3 | ρ⁺ Δatp9:: ARG8ᵐ | This study |

[a]*COX3* fused at the 3' end to a sequence coding for hemagglutinin followed by the protein C tag.
[b]*COX2* fused at the 3' end to a sequence-coding for seven histidines.
[c](See Fig 1A).
[d](See Fig 1B).
[e](See Fig 3A).
[f]pG71/ST9 consists of the integrative *TRP1* plasmid YIp349 with *COX6* fused at its 3' end to a sequence-coding for the hemagglutinin followed by the protein C tag.
[g]YIp349-COX6-3met-C#7 consists of the integrative *TRP1* plasmid YIp349 with *COX6* fused at its 3' end to a sequence coding for three methionine codons followed by the protein C tag.

**Table 2.** Sequences of PCR primers.

| Primer | Sequence |
|---|---|
| var1-1 | 5'-ggcgagctcggtaaatataatattaaagttaaattaaact-3' |
| var1-2 | 5'-ggcctgcagcctt-atttaataaaaagattataatctttatatatattaacc-3' |
| var1-3 | 5'-ggcagatcttatagttccccgaaaggag-3' |
| var1-5 | 5'- ggcggatccgctccaatatatttagctgctaataccaattgc-3' |
| cox6-11LF | 5'- ggccaattgtctgacgcacatg-atgaagaaac-3' |
| cox6-4LF | 5'-ggcggatccttacttaccatcgattaaccgtggatc tacctgatcttccatcatcatagaagagcttggaaataactcttcct-3' |
| arg8m-12 | 5'-ggccaattgaaaagatatttatcatcaacatcatc-3' |
| arg8m-2 | 5'-ggcggatccttaagcatatacagc-3' |
| cox6-51 | 5'-ggcgagctccatacgagccaatcag-3' |

The strain aMRS-sA9R (s stands for short ATP9 promoter) was obtained by a spontaneous deletion that occurred in the *ATP9* promoter region of aMRS-A9R leaving only 11 nucleotides right after the stop codon of *VAR1*, followed by 410 nucleotides upstream of the methionine initiation codon of *ATP9*. The deletion probably occurred because of the presence of an identical 44-nucleotide sequence (ATAGTTCCGGGGCCCGGCCACGGGAGCCG-GAACCCCGAAAGGAG) in both 5′*UTR* of ATP9 and 3′*UTR* of VAR1 that is only about 600 nucleotides away in the aMRS-A9R strain.

### Construction of strains that express mitochondrial *COX6-C*$^m$

To clone *COX6-C*$^m$ into a plasmid containing the *ATP9* promoter region, flanked by the 3′ UTR of *VAR1*, the latter was amplified from W303 total DNA with primers var1–1 and var1–2 and was then cloned into the YIp349 vector. The resulting pVAR1/ST1 plasmid was digested with BamHI and ligated to a BglII–BamHI fragment of the *ATP9* promoter region, which had been amplified with primers var1–3 and var1–5. The resulting plasmid pVAR1/ST7 was digested with MfeI and BamHI and ligated to an MfeI-BamHI fragment of *COX6* (pVAR1/ST8). *COX6* was amplified with primers cox4-11LF and cox6-4LF from plasmid pG71/ST20 containing the recoded sequence of *COX6* modified at the Leu73 and Leu143 codons (CTC to TTG and CTA to TTA, respectively) for expression in mitochondria (Macino et al, 1979; Bonitz et al, 1980) fused to three methionine codons at the C-terminus followed by the protein C tag. We have also omitted the native *COX6* presequence. The *COX6-C*$^m$ starts with the three initial codons of the *ATP9*-coding region followed by Ser41 of *COX6*. The SacI-PstI fragment of pVAR1/ST8 was transferred to pJM2 (Mulero & Fox, 1993), in which the PstI site on the 3′ side of *COX2* had been destroyed. The resultant plasmid, pVAR1/ST9, was introduced into the *kar1* mutant DFK$\rho^0$ by biolistic transformation (Bonnefoy & Fox, 2007). Transformants containing pVAR1/ST9 were screened by testing for respiratory competence of the diploid cells issued from a cross to the *cox2* tester aM9-94/A1. The synthetic petite DFK/VAR1/ST9 $\rho^-$ obtained from the transformation was crossed to aW303ΔCOX6$\rho^0$, a strain that harbors a *cox6* null allele and lacks mtDNA. Cells that had taken up the plasmid were screened by crosses to the mit$^-$ tester M9-94/A1. They were further crossed to aW303ΔCOX6$\rho^+$. Only diploid cells that underwent recombination of *COX6-C*$^m$ into the mitochondrial genome (a/αW303ΔCOX6/COX6-C$^m$) are able to respire. The haploid strain aW303ΔCOX6/COX6-C$^m$ was obtained by tetrad dissection after sporulation of the diploid strain.

### Construction of W303/COX6-C, a yeast strain with a nuclear gene that encodes Cox6 followed by three methionine codons and the protein C tag at the C-terminus

*COX6* was amplified with primers cox6-51 and cox6-4LF from total DNA of W303-1B. The PCR product was digested with SacI and BamHI and ligated to YIp349 digested with the same enzymes. The resulting plasmid YIp349-COX6-3met-C#7 was linearized with BstXI and transformed into strain W303ΔCOX6, yielding strain W303/COX6-C.

### Construction of aMRSΔATP9/COX6-HAC

The mitochondrial genome of aMRS/COX6-HAC (*arg8*) was deleted by incubation with ethidium bromide yielding aMRS/COX6-HAC $\rho^0$, which was then crossed to the *kar1* mutant DFKΔATP9 $\rho^+$ in which *ATP9* was replaced by *ARG8*$^m$. Arginine prototrophic transformants (aMRSΔATP9/COX6-HAC) were selected on a solid medium lacking arginine.

### Miscellaneous procedures

Purification, ligation, and transformation of DNA in *E. coli* were done under standard conditions (Green et al, 2012). Yeast was transformed by the lithium acetate method (Schiestl & Gietz, 1989). Western blots were treated with monoclonal or polyclonal antibodies followed by a second reaction with anti-mouse or anti-rabbit IgG conjugated to horseradish peroxidase (Sigma-Aldrich) and proteins detected with a SuperSignal chemiluminescent substrate kit (Pierce). Protein concentration was determined by the Folin procedure (Lowry et al, 1951).

## Supplementary Information

## Acknowledgements

This research was supported by the National Institutes of Health Grant 5RO1 GM111864 to A Tzagoloff, FAPESP Post-Doctoral Fellowship 2019/02799-2 and 2019/16015-3 to LVR Franco and FAPESP Grant 2020/05812-7 to MH Barros.

### Author Contributions

LVR Franco: conceptualization, data curation, formal analysis, validation, investigation, methodology, and writing—original draft, review, and editing.
C-H Su: investigation and methodology.
L Simas Teixeira: investigation.
J Almeida Clarck Chagas: investigation.
MH Barros: resources, data curation, formal analysis, supervision, funding acquisition, validation, investigation, methodology, project administration, and writing—review and editing.
A Tzagoloff: conceptualization, resources, data curation, formal analysis, supervision, funding acquisition, validation, investigation, visualization, methodology, project administration, and writing—original draft, review, and editing.

### Conflict of Interest Statement

The authors declare that they have no conflict of interest.

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
