## [Reviewer comments · Life Science Alliance]

Life Science Alliance

Allotopic expression of COX6 elucidates Atco driven co-assembly of cytochrome oxidase & ATP synthase

Leticia Veloso Ribeiro Franco, Chen Su, Lorisa Simas Teixeira, Jhulia A. C. Chagas, Mario Barros, and ALEXANDER TZAGOLOFF

DOI: <https://doi.org/10.26508/lsa.202301965>

Corresponding author(s): ALEXANDER TZAGOLOFF, Columbia University and Mario Barros, Universidade de São Paulo

Review Timeline:

Submission Date:	2023-02-01
Editorial Decision:	2023-03-20
Revision Received:	2023-06-16
Editorial Decision:	2023-07-16
Revision Received:	2023-08-03
Accepted:	2023-08-07

Scientific Editor: Novella Guidi

Transaction Report:

March 20, 2023

Re: Life Science Alliance manuscript #LSA-2023-01965-T

Dr. Leticia Veloso Ribeiro Franco
Instituto de Ciencias Biomedicas, Universidade de Sao Paulo
Brazil

Dear Dr. Veloso Ribeiro Franco,

Thank you for submitting your manuscript entitled "Allotopic expression of COX6 elucidates Atco driven co-assembly of cytochrome oxidase & ATP synthase" to Life Science Alliance. The manuscript was assessed by expert reviewers, whose comments are appended to this letter. We invite you to submit a revised manuscript addressing the Reviewer comments.

Thank you for this interesting contribution to Life Science Alliance. We are looking forward to receiving your revised manuscript.

Sincerely,

B. MANUSCRIPT ORGANIZATION AND FORMATTING:

Reviewer #1 (Comments to the Authors (Required)):

The manuscript entitled "Allotopic expression of COX6 elucidates Atco driven co-assembly of cytochrome oxidase & ATP synthase" by Ribeiro Franco et al. describes the establishment of a yeast strain, in which the gene encoding the Cox6 subunit of complex IV of the respiratory chain is transferred from the nucleus into the mitochondrial genome (mtDNA). This yeast strain is of interest, as it can be used to analyze protein (super) complex formation and turnover in the mitochondrial inner membrane, using radioactively labeled proteins. Whereas the description of strain construction and its biochemical properties is very well described, using a couple of sophisticated biochemical methods, the reviewer cannot identify the rationale behind the study, or a common thread within the paper. In addition, it remains unclear how the data presented can be concluded in the model presented in Figure 10. I cannot find appropriate controls. Which results within the paper contribute to the model?

Therefore, I cannot recommend publication of this study in the journal. Without having an idea what the authors intended to analyze, it does not make any sense to deeply review the paper.

Minor points:

- 1) The abstract is difficult to understand and needs revision for being more concise and informative.
- 2) For every result section, it remains unclear why the experiment has been done, and what the authors in the end learned from the experiment

Reviewer #2 (Comments to the Authors (Required)):

In this work, Franco and colleagues use yeast genetics and blue native gel electrophoresis to unravel a novel mechanism bridging COX and ATP synthase biogenesis via Atco, an assembly intermediate containing Cox6 and ATP9 subunits. The authors allotopically express Cox6 by introducing it into mtDNA under the control of the ATP9 promoter. This COX6m mutant was able to complement cox6-less yeast, although about 80% rho-0 mutants formed. S35 labeled of mitochondria confirmed expression of COX6m, but also revealed the presence of an additional band, possibly due to partial degradation of the protein, and of ATP9. BNGE experiments showed reduced amounts of COX and the presence of free bc1-complex, while ATP synthase was normal. Western blot immunovisualization revealed particularly levels of Cox1 protein, as expected as Cox6 is a component of Cox1 module. The Atco complex was analyzed by 2-D BNGE, which showed that most of the Cox6m band was not incorporated in the complex. Additional experiments showed that Cox6m was associated to the inner mitochondrial membrane, but not assembled with ATP9 in the Atco complex. Finally, by studying an ATP6-less strain, that authors were able to show that ATP synthase was necessary for COX assembly.

The paper is interesting and new. I have some comments though that should be addressed.

- 1) The first paragraph on the ARG8 mutants looks like a standalone and the rationale for including this part should be better explained. Also, the rationale for using the ATP9 promoter is unclear.
- 2) the pull-down experiment with anti-protein C beads shows a band interpreted as a degradation product of COX6m. However, the anti-Cox6 antibody does not recognize it. Obviously, this may be due to the absence of the epitope, but can the authors better comment on this?
- 3) I suggest the authors use more informative titles for each paragraph, briefly describing the findings more than just what has been done.

Reviewer #3 (Comments to the Authors (Required)):

The manuscript by Franco et al details attempts to express Cox6 from the mitochondrial genome and a role of ATP9 complexes in the assembly of cytochrome c oxidase. The experiments are primarily biochemical characterizations of the complexes formed in mitochondria visualized by S35 labelling of mitochondrial encoded components. The first objective (allotopic expression of Cox6) is almost an exercise in phenomenology. The justification is not clear as to what the reader will learn from the experimental set-up other than that it can be done. To enhance this section the authors should add a more detailed discussion of the evolutionary changes that have occurred since/as a consequence of moving to the nuclear genome. This would add interest and value to this section. Although that presumes that decreased incorporation of the mitochondrial encoded Cox6 is solely as a result of its primary sequence and not the promoter region. If this is solely mitochondrial stability in the allotopically expressing strain then this is less interesting. A section should be added that clarifies which option the authors are suggesting. The second objective of the manuscript is the description of a role for ATP synthase intermediates in cytochrome c oxidase assembly. This is

a novel and intriguing result. It would be beneficial to know if any of the COX1 assembly module components are detectable on the 2D gels. For example, can the authors detect Shy1, Coa1, Mss51 in SDS dimension. Especially given that the size of the identified band with a * is more consistent with Cox2 than Cox1. Therefore, to strengthen the conclusion that the Atp9 complex is a scaffold for cox1 intermediates and not just coincident with size additional evidence is needed to reinforce the model that is suggested in Figure 10.

Overall, the results are consistent with the model proposed and provide new insights into the assembly of cytochrome c oxidase. This reviewer suggest that the manuscript would be improved by clarifying the thought process behind the allotopic expression and the implications of the results for a broader audience. Additionally immunoblot analysis of the cytochrome c oxidase assembly factors to verify the status of Cox1 and those assembly factors is necessary to support the conclusions of the manuscript.

Reviewer #1 (Comments to the Authors (Required)):

The manuscript entitled "Allotopic expression of COX6 elucidates Atco driven co-assembly of cytochrome oxidase & ATP synthase" by Ribeiro Franco et al. describes the establishment of a yeast strain, in which the gene encoding the Cox6 subunit of complex IV of the respiratory chain is transferred from the nucleus into the mitochondrial genome (mtDNA). This yeast strain is of interest, as it can be used to analyze protein (super) complex formation and turnover in the mitochondrial inner membrane, using radioactively labeled proteins. Whereas the description of strain construction and its biochemical properties is very well described, using a couple of sophisticated biochemical methods, the reviewer cannot identify the rationale behind the study, or a common thread within the paper. In addition, it remains unclear how the data presented can be concluded in the model presented in Figure 10. I cannot find appropriate controls. Which results within the paper contribute to the model?

Therefore, I cannot recommend publication of this study in the journal. Without having an idea what the authors intended to analyze, it does not make any sense to deeply review the paper.

Minor points:

- 1) The abstract is difficult to understand and needs revision for being more concise and informative.
- 2) For every result section, it remains unclear why the experiment has been done, and what the authors in the end learned from the experiment

Answer:

In this revision we explain the rationale of our study that can now be found in the summary and in the introduction sections. We also tried to improve the diagram (Fig. 10) of our proposed model.

Our goal was to study the role of Atco complex, if any, in biogenesis of cytochrome oxidase. This complex, consisting of Cox6 and Atp9, was previously shown by *in organello* radiolabeling to be a source of Atp9 for ATP synthase assembly. Atco and assembly intermediates in general are present in very small amounts that can be detected by radiolabeling but not by methods involving immunological detection. Because Cox6 is a nuclear gene product, translated on cytoplasmic ribosomes, the pulse labeling approach was more challenging with this COX subunit. Whole cells labeling that would be required instead of *in organello* is problematic because of the background contributed by some 6,000 cytoplasmic proteins that would also be labeled in such experiments. For this reason we adopted a strategy involving relocation of *COX6* to the mitochondrial genome. This is more explicitly stated in the revised manuscript.

The strain harboring mitochondrially encoded Cox6 displays a slower respiratory growth due to lower levels of cytochrome oxidase than in the wild type control. Two-dimensional analysis PAGE has enabled us to show that most of the Cox6 expressed from the relocated mitochondrial gene is present as a free subunit and only a small fraction is associated with Atp9 in Atco. In this labeling experiment we cannot have a control with *COX6* expressed from the nuclear gene as it does not get radiolabel by *in organello* translation of isolated mitochondria.

Regarding the model, we showed that there is no cytochrome oxidase in the absence of Atp9 and consequently ATP synthase. Nonetheless, ATP synthase is assembled in the absence of cytochrome oxidase. This is why in our model Atco couples cytochrome oxidase biogenesis to that one of the ATP synthase and not vice-versa. *In organello* radiolabeling of the strain expressing mitochondrially encoded Cox6 shows that only a small fraction of the relocated Cox6 was able to form Atco, which can explain the very low levels of cytochrome oxidase compared to the wild type.

Minor points:

- 1) The new abstract was re-framed to better explain the rationale of our study.
- 2) We have added brief introductory statements (in red) in the result sections to help the reader understand why the experiment was done and what was drawn from the results. We have also changed the titles of the sections to make it clear what was achieved with each experiment.

Reviewer #2 (Comments to the Authors (Required)):

In this work, Franco and colleagues use yeast genetics and blue native gel electrophoresis to unravel a novel mechanism bridging COX and ATP synthase biogenesis via Atco, an assembly intermediate containing Cox6 and ATP9 subunits. The authors allotopically express Cox6 by introducing it into mtDNA under the control of the ATP9 promoter. This COX6m mutant was able to complement cox6-less yeast, although about 80% rho-/0 mutants formed. S35 labeled of mitochondria confirmed expression of COX6m, but also revealed the presence of an additional band, possibly due to partial degradation of the protein, and of ATP9. BNGE experiments showed reduced amounts of COX and the presence of free bc1-complex, while ATP synthase was normal. Western bolt immunovisualization revealed particularly levels of Cox1 protein, as expected as Cox6 is a component of Cox1 module. The Atco complex was analyzed by 2-D BNGE, which showed that most of the Cox6m band was not incorporated in the complex. Additional experiments showed that Cox6m was associated to the inner mitochondrial membrane, but not assembled with ATP9 in the Atco complex. Finally, by studying an ATP6-less strain, that authors were able to show that ATP synthase was necessary for COX assembly.

The paper is interesting and new. I have some comments though that should be addressed.

- 1) The first paragraph on the ARG8 mutants looks like a standalone and the rationale for including this part should be better explained. Also, the rationale for using the ATP9 promoter is unclear.
- 2) the pull-down experiment with anti-protein C beads shows a band interpreted as a degradation product of COX6m. However, the anti-Cox6 antibody does not recognize it. Obviously, this may be due to the absence of the epitope, but can the authors better comment on this?
- 3) I suggest the authors use more informative titles for each paragraph, briefly describing the findings more than just what has been done.

Answer:

- 1) Since *COX6* has never been relocated to the mitochondrial genome, we first tested our choice of the promoter for expressing the gene and the locus in mtDNA where to insert it, using the recoded *ARG8* allele (*ARG8^m*), which from previous reports is known to

express functional acetylornithine aminotransferase when relocated from nuclear to mitochondrial DNA. The *ATP9* promoter was used because it is a strong promoter. This information has been added to in the revision.

- 2) There are some possible explanations on why we do not detect the second Cox6 band in western blot. As indicated by the reviewer, it could be a proteolytic product that lost the protein C tag but more likely it undergoes more extensive proteolysis resulting in a decrease to levels detectable by radiolabeling but not by immunoblotting. This is now explained in the text.**
- 3) The titles of the results section have been changed to better indicate the rationale and purpose of the experiments in each section.**

Reviewer #3 (Comments to the Authors (Required)):

The manuscript by Franco et al details attempts to express Cox6 from the mitochondrial genome and a role of ATP9 complexes in the assembly of cytochrome c oxidase. The experiments are primarily biochemical characterizations of the complexes formed in mitochondria visualized by S35 labelling of mitochondrial encoded components. The first objective (allotopic expression of Cox6) is almost an exercise in phenomenology. The justification is not clear as to what the reader will learn from the experimental set-up other than that it can be done. To enhance this section the authors should add a more detailed discussion of the evolutionary changes that have occurred since/as a consequence of moving to the nuclear genome. This would add interest and value to this section. Although that presumes that decreased incorporation of the mitochondrial encoded Cox6 is solely as a result of its primary sequence and not the promoter region. If this is solely mitochondrial stability in the allotopically expressing strain then this is less interesting. A section should be added that clarifies which option the authors are suggesting. The second objective of the manuscript is the description of a role for ATP synthase intermediates in cytochrome c oxidase assembly. This is a novel and intriguing result. It would be beneficial to know if any of the COX1 assembly module components are detectable on the 2D gels. For example, can the authors detect Shy1, Coa1, Mss51 in SDS dimension. Especially given that the size of the identified band with a * is more consistent with Cox2 than Cox1. Therefore, to strengthen the conclusion that the Atp9 complex is a scaffold for cox1 intermediates and not just coincident with size additional evidence is needed to reinforce the model that is suggested in Figure 10.

Overall, the results are consistent with the model proposed and provide new insights into the assembly of cytochrome c oxidase. This reviewer suggest that the manuscript would be improved by clarifying the thought process behind the allotopic expression and the implications of the results for a broader audience. Additionally immunoblot analysis of the cytochrome c oxidase assembly factors to verify the status of Cox1 and those assembly factors is necessary to support the conclusions of the manuscript.

Answer:

As indicated in our response on this point to the first reviewer, the reason for relocating *COX6* to the mitochondrial genome was to ascertain by *in organello* radiolabeling if *Atco* is a source of Cox6 (obligatory or not) for cytochrome oxidase biogenesis. Assembly intermediates like *Atco* are present in very low amounts that can be detected by radiolabeling

and not by immunoblotting. At present, biogenesis of the mitochondrial respiratory complexes and of the ATP synthase are best studied by radiolabeling of isolated mitochondria and performing pulse and pulse-chase to establish precursor-product relationships. This method produces clean results free of background as there are only a few mitochondrial gene products. This is how *Atco* was discovered and shown to be a source of *Atp9* for ATP synthase assembly. *Atco* also contains *Cox6*, which is encoded by a nuclear gene. Whole cells labeling is challenging and presents technical problems since some 6,000 genes would be labeled, producing an interfering background of other labeled proteins even after purified from antibody against the tagged protein of interest. Our solution to this problem was to relocate *COX6* to mitochondrial DNA to enable *in organello* radiolabeling.

As requested by this reviewer, we have added a paragraph speculating on how the proposed mechanism for maintaining and adjusting the stoichiometry in response to metabolic demands may have been instituted in response to the evolutionary appearance of new subunit components of the respiratory and ATP synthase complexes during evolution of mitochondria.

Regarding the inefficient assembly of the mitochondrially encoded *Cox6*, we do not believe it to be a problem of the promoter for two reasons. First, the same promoter was used to express *ARG8^m*. Second, western blot showed that the steady-state levels of *Cox6* were only marginally lower than wild type. We also do not believe it is a problem in *COX6^m* sequence as the protein with the same sequence (*COX6* followed by three methionine codons and a protein C tag) except for a few residues at the N-terminus (poorly conserved among different yeast), when derived from a nuclear gene and imported from the cytoplasm fully complements the *cox6* mutant. More likely, the problem lies in the different location and topology of the imported and mitochondrially translated *Cox6*. Our reasoning on this point is now included in the discussion.

Unfortunately, the experiment suggested by the reviewer cannot be done. The intermediates seen on the 2D gels are transient complexes present in very low amounts that can be detected when mitochondrial gene products are radiolabeled. *Shy1*, *Mss51* and *Coa1* are translated on cytoplasmic ribosomes and therefore would not be labeled when isolated mitochondria are pulsed with ³⁵S-methionine. Studies of the components of the *Cox1* intermediate were done using strain constructs that expressed *Shy1*, *Mss51* and *Cox6* tagged with protein C epitope and purified on protein C antibody beads to access whether radiolabeled *Cox1* was pulled down along (*McStay, Su and Tzagoloff, 2013 Mol Biol Cell*). We know from such pull-down experiments of *Cox6-C* that this subunit is part of the *Cox1* intermediate. Because of the tag on *Cox6-C*, *Atco* is also pulled-down by the protein C antibody.

The reviewer is right in noting the incorrect position of *Cox1* in the 2D gel. The band marked as * was wrongly placed and the asterisk is now removed. We apologize for our mistake and thank the reviewer for careful reading of the manuscript.

We thank the reviewers for their critical reading of the manuscript. Their suggestions have greatly improved the description and rationale of this study, which we hope will be deemed acceptable for publication.

July 16, 2023

RE: Life Science Alliance Manuscript #LSA-2023-01965-TR

Dr. ALEXANDER A TZAGOLOFF
Columbia University
1212 Amsterdam Avenue
NEW YORK, NY 10027

Dear Dr. TZAGOLOFF,

Thank you for submitting your revised manuscript entitled "Allotopic expression of COX6 elucidates Atco driven co-assembly of cytochrome oxidase & ATP synthase". We would be happy to publish your paper in Life Science Alliance pending final revisions necessary to meet our formatting guidelines.

- please address the final Reviewer 3's minor point
- please add a Running Title and a Summary Blurb/Alternate Abstract to our system
- please add ORCID ID for the corresponding (and secondary corresponding) author--you should have received instructions on how to do so
- please add the Twitter handle of your host institute/organization as well as your own or/and one of the authors in our system
- please add a conflict of interest statement to your main manuscript text
- please add an Author Contributions section to your main manuscript text
- please cite Table 2 in the manuscript text accordingly
- please add callouts for Figures 10A-D and S1A-B to your main manuscript text;

A. FINAL FILES:

B. MANUSCRIPT ORGANIZATION AND FORMATTING:

Sincerely,

Reviewer #1 (Comments to the Authors (Required)):

The revised manuscript is a significant improvement over the original. It is clear, concise, and well-organized. The golden thread is easy to follow, and the message of the manuscript is clear. It is now clear, why Cox6 is expressed from the mitochondrial genome. The role of the ATCO complex and Atp9 in both COX and ATP synthase assembly is well-described, and the novel findings are well-presented and discussed. The model of the study is clearly illustrated (see Fig. 10). I believe that this manuscript would be of high interest to the broad readership of the journal. There, I can recommend its publication in the current form.

Reviewer #3 (Comments to the Authors (Required)):

The manuscript describes the allotopic expression of COX6 in the mitochondrial genome. This provides an interesting tool to probe the role the ATCO complex (a complex formed between COX6 and ATP9) on COX assembly. The authors have addresses all the points raised in the previous review regarding reasoning for study and have clarified the writing.

Minor point: In Figure 4 the authors labels the lanes at the bottom of the gel as digitonin extracted and PC elution. However the PC elution does not "extend" to the control lane where no Protein C labelled protein is expressed. If this is an oversight it should be corrected.

Dear Editor,

Please see our responses point by point **in red** below.

Sincerely,

Leticia V. R. Franco

-please address the final Reviewer 3's minor point

It has been corrected. We thank the reviewer for noticing our mistake. The label was wrong and we have corrected it, extending the PC elution to the control.

August 7, 2023

RE: Life Science Alliance Manuscript #LSA-2023-01965-TRR

Dr. ALEXANDER A TZAGOLOFF
Columbia University
Biological Sciences
1212 Amsterdam Avenue
NEW YORK, NY 10027

Dear Dr. TZAGOLOFF,

Thank you for submitting your Research Article entitled "Allotopic expression of COX6 elucidates Atco driven co-assembly of cytochrome oxidase & ATP synthase". It is a pleasure to let you know that your manuscript is now accepted for publication in Life Science Alliance. Congratulations on this interesting work.

DISTRIBUTION OF MATERIALS:

Again, congratulations on a very nice paper. I hope you found the review process to be constructive and are pleased with how the manuscript was handled editorially. We look forward to future exciting submissions from your lab.

Sincerely,
